# Targeted Metabolomics of Organophosphate Pesticides and Chemical Warfare Nerve Agent Simulants Using High- and Low-Dose Exposure in Human Liver Microsomes

**DOI:** 10.3390/metabo13040495

**Published:** 2023-03-29

**Authors:** Garima Agarwal, Hunter Tichenor, Sarah Roo, Thomas R. Lane, Sean Ekins, Craig A. McElroy

**Affiliations:** 1Division of Medicinal Chemistry and Pharmacognosy, College of Pharmacy, The Ohio State University, Columbus, OH 43210, USA; 2Department of Chemistry and Biochemistry, The Ohio State University, Columbus, OH 43210, USA; 3Collaborations Pharmaceutical Inc., Raleigh, NC 27606, USA

**Keywords:** LC-MS/MS, NMR, intrinsic clearance rate, cytochrome P450, metabolism, metabolite, toxicity

## Abstract

Our current understanding of organophosphorus agent (pesticides and chemical warfare nerve agents) metabolism in humans is limited to the general transformation by cytochrome P450 enzymes and, to some extent, by esterases and paraoxonases. The role of compound concentrations on the rate of clearance is not well established and is further explored in the current study. We discuss the metabolism of 56 diverse organophosphorus compounds (both pesticides and chemical warfare nerve agent simulants), many of which were explored at two variable dose regimens (high and low), determining their clearance rates (*Cl_int_*) in human liver microsomes. For compounds that were soluble at high concentrations, 1D-NMR, 31P, and MRM LC-MS/MS were used to calculate the *Cl_int_* and the identity of certain metabolites. The determined *Cl_int_* rates ranged from 0.001 to 2245.52 µL/min/mg of protein in the lower dose regimen and from 0.002 to 98.57 µL/min/mg of protein in the high dose regimen. Though direct equivalency between the two regimens was absent, we observed (1) both mono- and bi-phasic metabolism of the OPs and simulants in the microsomes. Compounds such as aspon and formothion exhibited biphasic decay at both high and low doses, suggesting either the involvement of multiple enzymes with different *K*_M_ or substrate/metabolite effects on the metabolism. (2) A second observation was that while some compounds, such as dibrom and merphos, demonstrated a biphasic decay curve at the lower concentrations, they exhibited only monophasic metabolism at the higher concentration, likely indicative of saturation of some metabolic enzymes. (3) Isomeric differences in metabolism (between Z- and E- isomers) were also observed. (4) Lastly, structural comparisons using examples of the oxon group over the original phosphorothioate OP are also discussed, along with the identification of some metabolites. This study provides initial data for the development of in silico metabolism models for OPs with broad applications.

## 1. Introduction

Pesticides are widely used in agriculture despite safety restrictions worldwide. One class of pesticides, organophosphorus compounds (OPs), are synthetic insecticides implemented for pest control and crop protection throughout the world. As a result of the widespread use of OPs in agriculture, an estimated 385 million accidental acute exposures to OPs occur every year, leading to around 11,000 fatalities [1]. Additionally, due to the ease of availability, OP pesticides account for approximately 14–20% of global suicides leading to 110,000–168,000 deaths [2,3,4]. OPs have also been optimized for toxicity towards humans for use as chemical warfare nerve agents (CWNAs), which are generally categorized into three different classes known as the G-series, V-series, and, most recently, the novichok agents or A-series [5,6]. The acute toxic effects of CWNA and pesticide poisoning are well established, with over 700 thousand cases annually [1]. The main mechanism of action in acute toxicity is phosphylation of the active site serine in the enzyme acetylcholinesterase (AChE), leading to several downstream effects, including cholinergic crisis and death, depending on exposure. The use of CWNAs serves as a concern for general human safety, as reflected by events such as the use of sarin in the Syrian Arab Republic during the ongoing conflict [7], but more prevalent is the non-lethal and long-term exposure of OPs, which can cause delayed neuropathy or long-lasting and cumulative effects, including depression, muscle control loss, Gulf War syndrome observed in Gulf War veterans [8,9], or aerotoxic syndrome [10]. Although the primary mechanism of toxicity is well understood, the metabolism of OPs, especially in humans, has not been investigated as well [11,12,13,14,15]. The involvement of cytochrome P450s (P450s) in OP metabolism has been noted through multiple metabolic studies as the primary metabolic enzymes contributing to the activation of phosphorothioate-containing OPs through a desulfuration reaction to form an active toxic oxon metabolite that contributes to AChE inhibition [16]. P450s have also been shown to contribute to the metabolic detoxification of OPs, with some compounds undergoing a dearylation reaction, resulting in a detoxified metabolite [17,18]. Additional evidence demonstrates that CYP2D6 may serve a direct role in OP metabolism that could potentially be exploited for treatments in the future [19]. Paraoxonase-1 (PON1) has also been shown to contribute to OP metabolism by facilitating the hydrolysis of organophosphorus ester bonds [20,21]. Nonetheless, a comprehensive investigation of the rate of metabolism, especially one comparing the rates for a structurally diverse set of OPs, has not been performed. Investigation of the rate of metabolism as it relates to structure and also determining the primary enzymes responsible for metabolism could lead to preventative treatments designed to deter activation metabolism of OPs or to enhance the metabolism into non-toxic metabolites for those that do not require activation [15].

The goal of the current study is to determine the metabolic clearance rates of a variety of structurally diverse OPs for both high- and low-dose exposure regimens in human liver microsomes (HLMs). HLMs allow for in vitro metabolic testing of OPs in biological conditions, which over time will metabolize OPs, providing a clearance rate [22]. This study utilizes both one-dimensional NMR (^1^H, ^31^P, and ^19^F for OPs that contain fluorine) to explore a high-dose exposure regimen as well as multiple reaction monitoring (MRM) LC-MS/MS to explore a lower dose exposure regimen to determine the rate of metabolism of a variety of OPs and identify some of the resultant metabolites. This represents an important first step to providing the data, enabling the development of computational models for the prediction of metabolic transformations of OPs and CWNAs in humans, and may eventually aid in the development of more broad-spectrum treatments for OP poisoning.

## 2. Materials and Methods

The NMR spectroscopic data were recorded on either a Bruker AVIII 400HD or Avance NEO 400 MHz NMR spectrometer, and the data were processed using TopSpin 3.2 (Bruker, Billerica, MA, USA) and MestReNova 14.2 (Mestrelab Research S.L., Compostela, Spain). A Thermo TSQ Quantiva with Vanquish UPLC (Thermo Fisher Scientific, Waltham, MA, USA) was used to obtain LC-MS/MS. The LC-MS/MS data were analyzed using Thermo Scientific Xcalibur 4.3. The clearance rates were calculated using GraphPad Prism 9.2.0. 

All organophosphorus compounds (OPs) were purchased in neat form from either LGC Standards (Manchester, NH, USA) or Sigma-Aldrich (Milwaukee, WI, USA); reduced nicotinamide adenine dinucleotide phosphate (NADPH) tetrasodium salt (CAS 2646-71-1) was from Sigma Aldrich (Milwaukee, WI, USA), deuterated tris(hydroxymethyl)aminomethane (TRIS, CAS 202656-13-1), deuterated water and deuterated hydrochloric acid (DCl) were from Cambridge Isotope Laboratories (Tewksbury, MA, USA), TRIS base (CAS 77-86-1) was from Fisher Scientific (Florence, KY, USA), and ammonium formate (CAS 540-59-2) was from Sigma Aldrich (Milwaukee, WI, USA). Isotopically labeled OPs, acephate-*acetyl*-d_3_ (Ace-d3), chlorpyrifos-*diethyl*-d_10_ (CPy-d10), paraoxon-*ethyl*-d_10_ (PE-d10), and trichlorfon-*dimethyl*-d (TCP-d6) were also purchased from Sigma Aldrich (Milwaukee, WI, USA). Chemical warfare nerve agent simulants, 3-cyano-4-methyl-2-oxo-2H-chromen-7-yl ethyl methylphosphonate (EMP, coumarin simulant of VX), 3-cyano-4-methyl-2-oxo-2H-chromen-7-yl cyclohexyl methylphosphonate (CMP, coumarin simulant of cyclosarin), 3-cyano-4-methyl-2-oxo-2H-chromen-7-yl (3,3-dimethylbutan-2-yl) methylphosphonate (PiMP, coumarin simulant of soman), and ethyl (4-nitrophenyl) dimethylphosphoramidate (NEDPA, *p*-nitrophenol simulant of tabun) were generously provided by Dr. Christopher Hadad in the Department of Chemistry and Biochemistry at OSU, where they were synthesized using previously published procedures [23]. Mixed-gender human liver microsomes (HLMs, 150-donor pool, lot number ZZQ) were purchased from BioIVT (Hicksville, NY, USA). Methanol, acetonitrile, and formic acid (Fisher Scientific, Florence, KY, USA) were all high-performance liquid chromatography (HPLC) grade.

Stock solutions of OPs and CWNA simulants (120 mM, 5 mM, and 250 µM) were prepared in acetonitrile (AcN) and stored at −80 °C when not in use. Stock solutions of NADPH (20 mM) and deuterated TRIS (120 mM, pH = 7 using deuterated HCl) were prepared in D_2_O for NMR or in double-distilled water for LC-MS studies, both stored at −80 °C while not in use. Stock solutions of the internal standards (ISTDs: Ace-d3, CPy-d10, PE-d10, and TCP-d6) were prepared in AcN at 35.8 mM, 9.2 mM, 11.7 mM, and 25 mM, respectively. Further, a working solution of 10 ppm of each ISTD was prepared in AcN, and all ISTD solutions were stored at −80 °C when not in use.

For the high dose (NMR) studies, OPs (5 mM) were incubated with HLMs (0.5 mg/mL), deuterated TRIS buffer (10 mM), and NADPH (6 mM) in deuterated H_2_O (in a 750 µL total sample volume) before collecting ^1^H-, ^31^P-, and ^19^F- (based on structure) NMR data across a minimum of 2.5 days (or until fully metabolized) with spectra collected as fast as possible for those cleared most rapidly or every two hours for most OPs. The concentration of the HLMs for the study, 0.5 mg/mL, was chosen based on previously published literature [24,25,26]. This was followed by the addition of 1.5 mL of AcN to quench the reactions, after which they were stored at −80 °C for future evaluation. The NMR spectroscopic data were recorded on a Bruker AVANCE-III HD or Avance NEO 400 MHz NMR spectrometer, and the data were processed using TopSpin 3.2 (Bruker, Billerica, MA, USA) and MestReNova 14.2 (Mestrelab Research S.L., Compostela, Spain) software. ^31^P data were collected using the zgpg30 pulse program, which uses a phosphorus pulse flip angle of 30° and power-gated composite pulse decoupling of ^1^H using the WALTZ16 decoupling sequence with data collected with a spectral width of 406 PPM, a center at −50 PPM, 32 K complex points, and 128 scans. ^19^F data were collected using the zgig pulse program, which uses inverse gated composite pulse decoupling of ^1^H using the WALTZ16 decoupling sequence with data collected with a spectral width of 241 PPM, a center at −100 PPM, 64 K complex points, and 128 scans. ^1^H data were collected using the zg30 pulse program, which uses a proton pulse flip angle of 30° with data collected with a spectral width of 20 PPM, a center at 6.18 PPM, 32 K complex points, and 128 scans. Peak areas in the ^31^P NMR spectra were used to estimate the concentrations of OPs (and metabolites) at each time point, which were further applied to calculate the elimination rate constant (*k*) and intrinsic clearance rates (*Cl_int_*), as explained below. Additionally, the NMR data were analyzed to identify the metabolites being formed during the experiment to enable the prediction of the various metabolic pathways the compounds could undergo.

In the low-dose (LC-MS/MS) studies, OPs were separated into seven groups based on their literature LD_50_ values [27,28,29,30], with eight compounds in each set. The LD_50_ values of the simulants were estimated based on their comparison with the corresponding nerve agents [23,31,32,33,34,35]. The experimental setup was comparable to the high-dose study. Briefly, the OPs (2 * × LD_50_) were incubated with the HLMs (0.5 mg/mL), TRIS buffer (10 mM), and NADPH (1 mM) in 1 mL of H_2_O at 37 °C. Eleven time points (0, 10, 20, 30, 60, 120, 240, 360, 480, 720, and 1440 min) were collected by taking 75 µL of the reaction mixture and halting metabolism by adding AcN spiked with ISTDs (Ace-d3 (125 ng/mL), CPy-d10 (125 ng/mL), PE-d10 (75 ng/mL), and TCP-d6 (100 ng/mL)) and storing the samples at −20 °C until further analysis. After the last time point, all samples along with pooled QCs (low, mid, and high concentrations) and blanks were centrifuged at 1147× *g* for 30 min at 4 °C after which the supernatant was removed from the mixture, diluted in H_2_O, and vortexed for 10 min. A fresh batch of calibration standards was prepared for each group and processed in the same manner as described above. The time point samples, QCs, blanks, and calibration standards were subjected to LC-MS/MS analysis on a reversed-phase UPLC column (CSH Fluoro-Phenyl column, 1.7 µm particle size, 2.1 mm X 100 mm, Waters, Milford, MA, USA) on a TSQ Quantiva with a Vanquish UPLC as the front end. All compounds were detected in positive ion mode using the multiple-reaction monitoring (MRM) setting. A standard solution of 1 µg/mL, in acetonitrile, for individual OPs was directly infused into the electrospray ionization source of the MS system while scanning for protonated, sodiated, or ammoniated ions prior to mass spectrometric parameter optimization. Next, the optimal collision energies (CEs) and three most intense product ions were selected for each analyte, where the most intense product ion was used as the quantifier ion (Pdt1), and the other two product ions (Pdt 2 and Pdt 3) were used as qualifier ions to ensure that the peak being quantified was indeed the analyte of interest (shown in Appendix A). Quantification of each OP was performed using the linear range of a calibration curve consisting of fifteen different concentration levels ranging from 0.008 to 6000 ng/mL (shown in Appendix A). Water with 2.5 mM ammonium formate and 0.05% formic acid (FA) and methanol with 2.5 mM ammonium formate and 0.05% FA were used as mobile phases A and B, respectively, for the gradient elution. For all compounds, the gradient consisted of 0 to 2.5 min with 99% A/1% B; 2.5 to 2.51 min 55% A/45% B; 2.51 to 7.51 min linear gradient to 20% A/80% B; 7.51 to 8.5 min 20% A/80% B; 8.5 to 9 min linear gradient to 2% A/98% B; 9 to 12 min 2% A/98% B. The concentration of each OP at each time point was calculated using the calibration curves (fitted to a log–log line with R^2^ = 0.9999 with area ratio versus compound concentration; area ratio for the calibration curve calculated using the ratio of compound area and ISTD area; ISTD selected based on the nearest retention time). The concentration was then used to calculate the elimination rate constant (*k*, min^−1^) by plotting the concentration versus time and fitting it to a non-linear least square regression analysis for both a monophasic and biphasic decay model with the best-fit model selected using the extra sum-of-squares F-test whereby the simpler model was selected unless the *p*-value was less than 0.05 (NMR and MS data, GraphPad Prism 9.2.0, criteria: probability of *p* to enter was ≤ 0.05, span (Y0-plateau for first order decay or (Y0-Plateau) × PercentFast × 0.01 for second-order) ≥ 2 × standard deviation at 95% confidence interval). The following equations were used to calculate the in vitro half-life (*t_1/2_*, min) and intrinsic clearance rate for HLMs (*Cl_int_*, µL/min/mg of protein), where *V* represents the volume of incubation in µL, and *A* is the amount of protein added during the incubation in mg.
(1)t12=ln2k
(2)Clint=ln2×V t12×A

## 3. Results and Discussion

The clearance rate comparison of the high and low-dose regimen studies is summarized in Figure 1. There were seven compounds of the 56 studied that failed the clearance criteria of the concentration vs. time graph, having a span ≥ 2 times the standard deviation of the concentration measurements, and hence their clearance rates were considered negligible. For the remaining compounds, most OPs followed a monophasic decay in both the high and low-dose regimens. Since the clearance rate of any drug is dependent on the half-life and the elimination rate constant, which is also dependent on the concentration of the drug at a given time in the plasma (or in this case microsomal solution, $C_{t}=C_{0}.e^{-kt}$), it is expected that the clearance rates between the two studies would not be exactly the same.

Only two OPs, formothion (**1**, Figure 2) and aspon (**5**, Figure 2), demonstrated biphasic metabolism at both dose regimens, with 11 others demonstrating biphasic metabolism in one of the two-dose regimens (Appendix A). The fast clearance rate for four compounds, chlorpyrifos oxon (CPO), diazinon, fenthion, and isofenphos (**2**, Figure 2), was too fast to yield enough points among the selected sample time points to allow for appropriate fitting; thus, only the slower decay is reported. In line with some literature, one possible explanation for the biphasic effect is that one of the metabolites is either a substrate (leading to competitive inhibition) or inhibitor of the same P450 that is metabolizing the OP, thereby inhibiting the original metabolism [36]. Another plausible explanation could be that there are multiple P450s acting on the same compound but with different k_cat_ and *K*_M_ values. In this scenario, likely the fast clearance rate would be due to an enzyme with both a higher k_cat_ and *K*_M_, thereby leading to fast clearance until the concentration of the OP is low enough to no longer bind to the enzyme (due to its high *K*_M_); then, the remaining compound would be broken down by another P450 with lower k_cat_ and *K*_M_ (making it able to bind and turnover the OP at the lower concentrations but at a slower rate).

In the case of formothion (**1**, Figure 2), fast and slower clearance rates of 22.27 µL/min/mg and 2.623 µL/min/mg were observed in the high-dose studies, which was comparable to the rates in the low-dose studies of 24.56 µL/min/mg and 3.50 µL/min/mg. Based on the ^31^P NMR shifts, it was evident that one of the major metabolites formed was another OP being studied, dimethoate (**3**, Figure 2), *δ_P_* 97.61 ppm (Appendix A), which was likely formed by enzymatic cleavage or hydrolysis and is not metabolized further by HLMs as it is one of the slowly metabolized compounds at both 5 mM and 33.95 µM. However, a comparison of the phosphorus spectra of formothion in the presence and absence of HLMs suggested that all the metabolites observed in the former study were possibly due to hydrolysis as they were present in the latter as well. Another explanation is that it may be due to the breakdown of NADPH, although this would require further exploration. There was one metabolite peak, though minor and unidentified, that was observed at *δ_P_* 0.92 ppm in the microsomal incubation experiment that was absent in the hydrolysis study, suggesting this could be a product of biotransformation.

Among the remaining OPs that demonstrated biphasic metabolism (Appendix A), there was an even split between the study groups, i.e., four compounds showed biphasic curves in NMR but not in the MS studies and vice-versa. For example, malathion (**4**, Figure 2), a broad-spectrum phosphorothioate, was one of the fastest metabolized compounds with a fast clearance rate of 2245.5 µL/min/mg and a slower rate of 10.09 µL/min/mg at a lower dose of 0.17 mM, but only a monophasic metabolism of 11.15 µL/min/mg at the higher 5 mM concentration. Further exploration of the ^31^P NMR suggested fast metabolism of malathion, *δ*_P_ 95.89 ppm, occurs within the first 2 h, and the metabolite formed at *δ*_P_ 95.02 ppm (Appendix A) further breaks down to give some or all the other metabolites observed while exhibiting a biphasic decay at this higher concentration. The major metabolite for malathion was observed at *δ*_P_ 97.64 ppm, corresponding to *O*, *O*-dimethyl dithiophosphate [37]. However, this observation is in contrast to previous studies of malathion 4, where malaoxon (signal around 28.31 ppm) [38], a more toxic and potent inhibitor of AChE or dimethyl thiophosphate (expected at around *δ*_P_ 27 ppm) [39], were shown to be the major metabolites of malathion but were absent after three days of HLM incubation (Appendix A). Several published biotransformation studies of malathion have shown that the formation of malaoxon is dependent on the concentration of the CYP2C P450 family in the HLMs [38,40,41]. Since it was not clear what concentrations of the different CYP2C enzymes (CYP2C8, CYP2C9, and CYP2C19) were present in the microsomes used for this study, this could be a possible explanation for the lack of formation of the toxic metabolite of malathion [37,40]. Consequently, it is difficult to establish a clear role for P450s in metabolizing malathion without further investigating the quantification of individual P450 isoforms in the HLMs along with data-independent mass spectrometric analysis of these reactions. 

The major challenge for the high-dose studies of the OPs was the solubility of the OPs at higher concentrations in water. While the simulants were readily water soluble at high concentrations, many of the OPs had low or no solubility in water at the 5 mM concentration. Therefore, only 30 of the compounds could be studied in the high-dose studies by NMR, compared to the 56 compounds studied in the low-dose regimen by LC-MS/MS. Two other major differences between the two dose regimens were the duration and temperature of the metabolism studies; for NMR, we collected data for a minimum of 2.5 days, and samples had to be maintained at room temperature as they were kept in an autosampler that lacks temperature control (some compounds that did not show any difference after that were continued for two more days and then arrested) whereas the LC-MS/MS studies were over a period of 24 h and samples were maintained at 37 °C. Despite the limited number of compounds and time and temperature differences, the high-dose data were insightful. Of the 30 OPs, only 13 of them were metabolized to less than 30% remaining, and of these 13, 4 compounds demonstrated biphasic metabolism: aspon (**5**, Figure 2), EMP (**6**, Figure 2), TCP (**7**, Figure 2), and malathion (**4**, Figure 2). EMP (**6**, Figure 2), a coumarin analog of VX, was the only nerve agent simulant that demonstrated a biphasic metabolism at higher concentrations. The major peak, observed at *δ*_P_ 26.74 ppm (Figure 3), corresponds to a common metabolite of all of the nerve agent surrogates, ethyl methylphosphonate, which is formed by cleavage of the P-O bond on the coumarin [42,43], and contributed to the fast half-life of EMP with a 5.59 µL/min/mg clearance rate. The P-O bond cleavage may be due to hydrolysis or could be enhanced by paraoxonases or P450 enzymes, such as CYP1A2, CYP3A4, and CYP2C19, which perform dearylation reactions in humans [15,21,44]. In the ^31^P NMR spectra of EMP collected in the absence of NADPH, although peaks at *δ*_P_ 26.74 ppm and 35.76 ppm were observed, they did not increase over the 24 h period, suggesting enzymatic transformations of EMP. Another easily identified metabolite observed at *δ*_p_ 30.55 ppm, corresponding to methyl phosphonic acid [45], could be formed by two pathways: either from ethyl methyl phosphonic acid or another metabolite at *δ*_P_ 32.71 ppm (Figure 3). Five other metabolites were observed at *δ*_P_ 35.76, 32.71, 32.64, 32.62, and 1.52 as minor metabolites in the ^31^P NMR spectra (shown in Figure 3). Further, the peaks at *δ*_P_ 35.76 (unknown metabolite) and 32.71 ppm 3-cyano-4-methyl-2-oxochromen-7-yl-oxy-methylphosphonate (or metabolite 2, Figure 3) show an increase in concentration for the first 100 min but decreasing concentration thereafter, suggesting that these metabolites are also potential substrates for the P450s or other enzymes present in the microsomes. Due to the lack of more informative data, such as 2D-NMR or high-resolution mass spectrometry, it is difficult to determine the identity of all the metabolites in this study, although they could be due to modifications on the coumarin moiety of the compound.

^31^P NMR was very useful in identifying some expected metabolites from OP incubations with HLMs, such as diethyl phosphonic acid (*δ*_P_ 0.77 ppm, Appendix A), dimethyl phosphonic acid (*δ*_P_ 3.02 ppm, Appendix A), *O*, *O*-dimethyl dithiophosphate (*δ*_P_ 97.64 ppm, Appendix A) and phosphoric acid. In some cases, such as mevinphos (MVP, **8**, Figure 2) and phosphamidon (PPM, **9**, Figure 2), both NMR and MS identified the presence of configurational isomers, and in each case, the *Z* and *E* isomers demonstrated a different metabolic rate at the low and high doses. These isomers were present in an approximately 2:1 ratio (determined based on the integration of the ^31^P NMR signals), which was also consistent with the ratio of the peak areas of the protonated species in LC-MS/MS data. It was difficult to assign which isomer corresponded to which peak; however, the phosphorus chemical shifts and retention times were only sufficient to differentiate the isomers, as I (larger) and II (smaller) for both OPs. Interestingly, both MVP-I and II were part of the group that did not meet the clearance criteria (span ≥ 2 × std dev) at the 5 mM concentration, but at 0.50 µM, both isomers were depleted to < 1% in the microsomes. This could possibly be due to the lower concentration being closer to the k_cat_/*K*_M_ values of the P450 enzymes, while the higher concentration of PPM inhibits the enzymes, thereby preventing any metabolism of the OP [36]. We found no literature to support or refute these results. In contrast, in the case of PPM-I and II, we observed a monophasic metabolism at both the high and low doses, but the rates were different for the *Z* and *E* isomers (observed in a 2.1:1 ratio in both ^31^P NMR and mass spec). Based on the chemical shifts and retention time, only 20% of PPM-I (potentially the *Z* isomer, the higher peak area assigned based on literature) [28] was metabolized at both dose levels compared to PPM-II, which was depleted to 20% at 1.42 µM but only to 60% at the 5 mM concentration. This is a good indication that isomeric differences affect the rate of metabolism in addition to the OP concentration or dose. It would be interesting to determine which isomer of PPM leads to toxicity in mammals which is beyond the scope of this study.

The biggest advantage of conducting the low-dose studies was the ability to study so many different OPs—56 characterized for this study. This enabled us to not only examine a set of structurally diverse compounds, but also to compare those that had a single functional group difference. One example discussed above is formothion and dimethoate, which differ by an aldehyde group, where the former is metabolized, but the latter is not depleted at all. Another such structural variation is the phosphorothioate versus the oxon metabolite, with the example of chlorpyrifos (CPy, **10**, Figure 2) and chlorpyrifos oxon (CPO, **11**, Figure 2). Based on the MS data, both CPO (5.98 µM) and CPy are metabolized (6.57 µM) to 0.5% and 28%, respectively. It was evident that CPO had a faster metabolic rate than CPy, which shows the formation of the former as a metabolite (Figure 4), consistent with the literature [16,46]. CPy is one of the few pesticides for which the metabolism is well studied, and several schemes in the presence of P450s have been proposed [46,47,48,49,50]. Other compounds with a similar metabolism, such as leptophos (**12**, Figure 2) and leptophos oxon (**13**, Figure 2), also demonstrated the phenomenon where the presence of the oxon metabolite was observed in the phosphorothioate metabolism by the HLMs, as shown in Figure 4. Due to low or no solubility in deuterated H_2_O, these compounds were not studied at the high-dose regimen, except for CPO, which showed very fast metabolism at the high concentration of 5 mM (Appendix A). Conversely, based on the absolute quantification of the OPs in HLMs after 24 h incubation, the oxon metabolite was observed to be only a small percent (2% CPy in CPO and 18% leptophos oxon in leptophos) of the parent organophosphate (panel B in Figure 4), suggesting that the oxon form is definitely a metabolite of the phosphorothioate, but not the only or most abundant metabolite, thereby necessitating further exploration of the metabolites formed.

## 4. Conclusions

Our current understanding of neurotoxic organophosphorus agents is limited to metabolism by cytochrome P450 enzymes and, to some extent, metabolism by esterases and paraoxonases in humans. The role of compound concentrations on metabolite formation and clearance by these enzymes has not been well established and is further explored in the current study. Herein, we described the metabolism of 56 diverse OPs (pesticides and nerve agent simulants), many at two variable doses, high (5 mM) and low (2 × LD_50_), by their clearance rates (*Cl_int_*) in human liver microsomes. Both 1D-NMR and MRM LC-MS/MS were used to calculate the *Cl_int_*, which ranges from 0.001 to 2245.52 µL/min/mg of protein at the LD_50_ level and from 0.002 to 98.57 µL/min/mg of protein at the high dose regimen. Moreover, comparing the two different dose regimens enabled us to not only identify several metabolites, such as dimethyl dithiophosphonate, diethyl and dimethyl phosphonic acid, but also demonstrated the impact of structural modifications on the rate of metabolism. More importantly, the study of different concentrations allowed us to focus on the effect of concentration on the clearance rates allowing speculation on the role of *K*_M_ and k_cat_ in OP degradation in HLMs. Another key observation was the ability of multiple substrates, in this case, metabolites and OPs, to act on the same or different enzymes. Though the study was insufficient to comprehensively describe the metabolism pathways and enzymes involved, it is an important first step towards exploring this area and understanding the substrate–enzyme interactions. Future studies using untargeted LC-MS/MS analysis, along with 2D-NMR experiments, will be beneficial for identifying the metabolites and elucidating the pathways and enzymes involved in the biotransformation of these OPs in HLMs and/or hepatocytes by the P450s. This study provides preliminary data to enable the development of in silico models for the metabolism of OPs to predict the P450s involved in the metabolism of OPs, their clearance rates, likely metabolites, and possible metabolic pathways for newly emerging OP threats.

## Figures and Tables

**Figure 1 metabolites-13-00495-f001:**
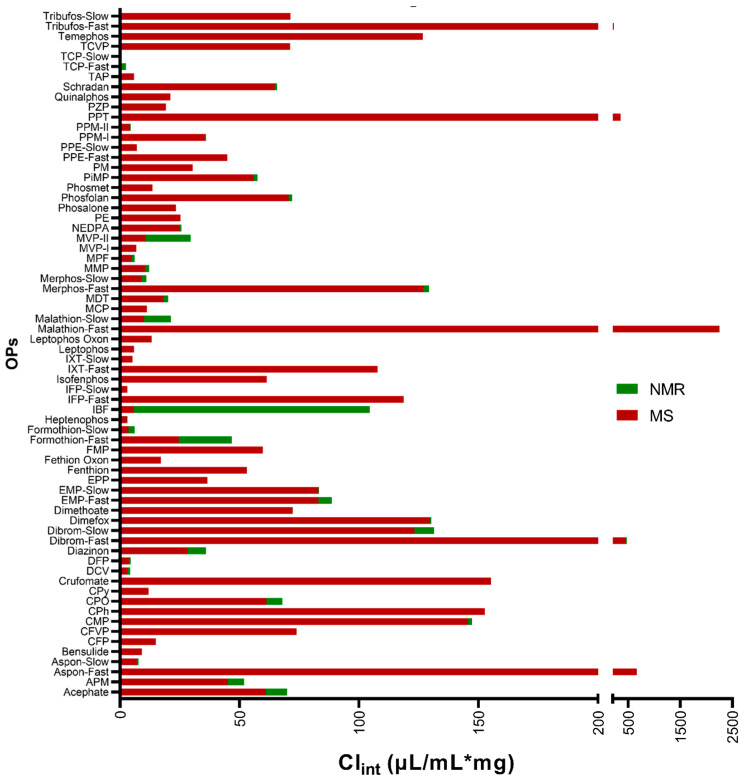
Comparison of the intrinsic clearance rates of OPs in HLMs calculated from data obtained at low- and high-dose regimen experiments.

**Figure 2 metabolites-13-00495-f002:**
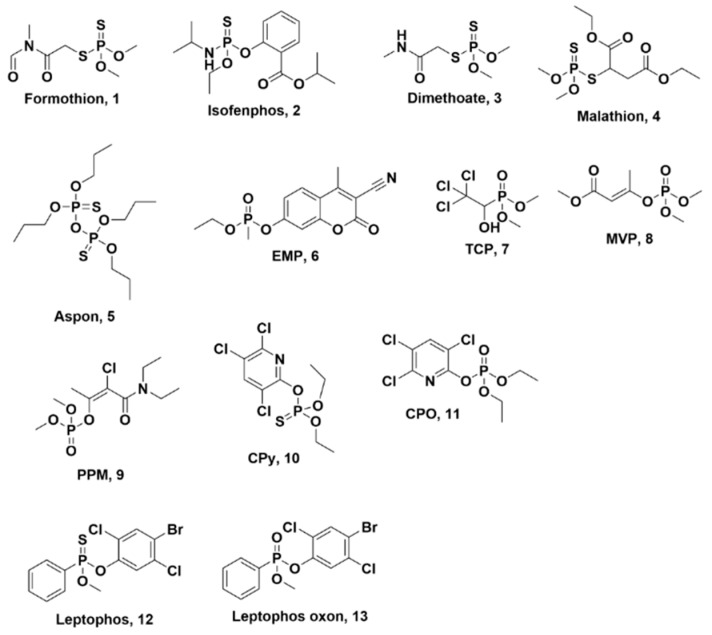
Structures of select OPs used in HLM metabolism study.

**Figure 3 metabolites-13-00495-f003:**
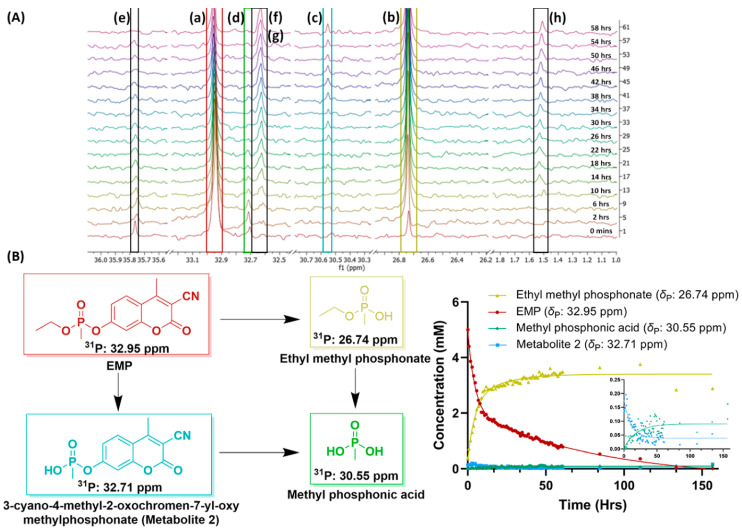
Top Panel: (**A**) Stacked ^31^P-NMR spectra of EMP (6) in HLM over a period of 3 days, identifying its metabolites and their chemical shifts; (a) EMP chemical shift, *δ*_P_ 32.95 ppm; (b) major metabolite of EMP, ethyl methylphosphonate at *δ*_P_ 26.74 ppm; (c) *δ*_P_ 30.55 ppm corresponding to methyl phosphonic acid; (d) metabolite of EMP and substrate for CYPs at *δ*_P_ 32.76 ppm; (e–h) minor metabolites of EMP. Bottom Panel: (**B**) Structures of the major metabolites of EMP (4) in the presence of HLMs identified using ^31^P-NMR (in D_2_O; left) and the curves corresponding to the integration of each peak in the ^31^P spectrum at each time point (top right) with an expansion of the metabolites with the lowest concentrations (bottom right).

**Figure 4 metabolites-13-00495-f004:**
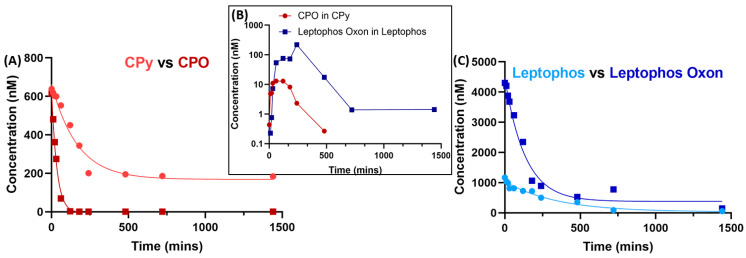
(**A**) Metabolism of chlorpyrifos (CPy) and chlorpyrifos oxon (CPO) in HLM at 2 × LD_50_ concentration over 24 h at 2 × LD_50_ concentration. (**B**) Graph showing the appearance and disappearance of the oxon metabolites (leptophos oxon is blue and CPO is red) detected in the runs of the phosphorothioate parent OPs during the lower dose mass spectrometric experiments. (**C**) Metabolism of leptophos and leptophos oxon in HLM at 2 × LD_50_ concentration over 24 h at 2 × LD_50_ concentration.

## Data Availability

The data that support the findings of this study are available from the corresponding author upon request. Data is not publicly available due to privacy.

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
