# Peer review of "Targeted Metabolomics of Organophosphate Pesticides and Chemical Warfare Nerve Agent Simulants Using High- and Low-Dose Exposure in Human Liver Microsomes"

_metabolites, 2023, doi:10.3390/metabo13040495_

Round 1

Reviewer 1 Report

This manuscript is an intriguing piece of work that explores the metabolism of 56 organophosphorus pesticides and chemical warfare nerve agent simulants under two different dose regimens in human liver microsomes. Although this is preliminary data, I found the paper as another step to reduce newly emerging OP threats. I would like to offer the following comments for the authors' consideration:

1.    With regard to the dose regimens, what was the rationale behind using 5mM as a high dose? Some of the OPs had low or no solubility at 5 mM and may not reach such a high concentration in vivo.

2.    To determine the metabolism rate of OPs, the authors used LC-MS/MS for low dose regiment and NMR for high dose regiment, respectively. Have the authors compared the accuracy and precision of the two methods?

3.    Can the authors provide more details on the parameters used for recording NMR spectra?

4.    In this study, 1H-, 31P- and 19F-NMR data were collected. But only 31P-NMR data were presented in the paper.

5.    What was the incubation volume and incubation time for the high dose studies?

6.    Is 20.04 µL/min/mg at line 235 a typographical error?

Reviewer 2 Report

The authors described the metabolism of 56 diverse organophosphates by their clearance rates in human liver microsomes. Even though future studies are required to provide further understanding on the exact mechanisms of these organophosphates, this article provides a good fundamental for future research. The discussion is well written and the limitations of the studies is described in the conclusion. 

Author Response

The authors thank the reviewer for their assessment of this manuscript. We definitely agree that this is just the first step. Our future work (provided we can secure additional funding) will be to follow up these studies with high resolution mass spectrometric studies using a semi-targeted and untargeted workflow to identify the metabolites that are being formed as well as incubation with recombinant CYP450s to determine which enzymes are responsible.

Reviewer 3 Report

The topic is very technical and specific, hence it will probably attract a limited number of readers. Still, I found it interesting. I also liked the use of human liver microsomes instead of any animal model. 

Author Response

The authors thank the reviewer for their comments and their detailed reading of the manuscript. We agree that this is a technical and specific topic, but hope that other readers will find the results interesting just as the reviewer did. We would also like to thank the reviewer for providing keen insight into strategies to increase the readership through additional keywords (which have been added to the revised version) as well as catching the omission of the publication date of one of the references (which has also been added to the revised version).

Reviewer 4 Report

The work of Agrawal and Al dealt with the identification of metabolites of phosphorus pesticides by low resolution LC-MS/MS and by NMR. the work is well presented and clearly targets the limitations of this study. the main shortcoming of this study lies in the fact that the identification of smetabolites is only made by low resolution and targeted mass spectrometry. it would have been profitable to use high resolution mass spectrometry to follow the metabolization of the compounds.

Author Response

The authors would like to thank the reviewer for the insightful comments. First, we would like to emphasize that the metabolite identifications included in this manuscript were made using the NMR data which provided the structural information required for appropriate identification. The only metabolite formation that we followed using the low resolution mass spectrometry was the formation of the oxones from the corresponding thiones in the cases where we had developed the methods for both. Nonetheless, the reviewer's thoughts are exactly in line with ours... The current manuscript that utilized a targeted approach was the beginning work where we determined the clearance rates for a large number of OPs. Our future plans (provided we can secure further funding) are to use high resolution mass spectrometry with semi-targeted and untargeted workflows to identify all of the metabolites formed during the metabolism of the OPs that demonstrated the fastest clearance rates.